# Influence of study shift on the interrelationships among chronobiological factors, health practices, and anthropometry in adolescents

Raphael Corrêa Martins[1], Flávia dos Santos Barbosa Brito[2], Cintia Chaves Curioni[3]*

**1** Federal Center for Technological Education Celso Suckow da Fonseca, Rio de Janeiro, Brazil, **2** Postgraduate Program in Food, Nutrition and Health (PPG-ANS), State University of Rio de Janeiro, Rio de Janeiro, Brazil, **3** Department of Social Nutrition – Institute of Nutrition, State University of Rio de Janeiro, Rio de Janeiro, Brazil

\* cintiacurioni@gmail.com

## Abstract

The mismatch between chronobiological predispositions and social demands makes it difficult for adolescents with an evening chronotype to maintain healthy habits. This study aimed to explore the interrelationships among chronobiological factors, health practices, and anthropometry in adolescents attending morning and afternoon class shifts. In this cross-sectional study, 925 adolescents (14–19 years old) completed an online questionnaire covering food practices, chronotype, sleep duration, social jetlag, screen time, socioeconomic data, physical activity, school shift, and anthropometric measurements. The interrelationships among these variables were analyzed through structural equation modeling using six path models. Food practices were categorized into "complete meals" (breakfast, lunch, dinner, and bean consumption) – as healthy practices, "unhealthy foods" (sweets, fried snacks, soft drinks, salty ultra-processed foods, and fast food) and "vegetables and fruits," stratified by shift (morning or afternoon), with body mass index by age (BMI/age) as the main outcome. Among students in the morning shift, reduced sleep time was directly associated with increased BMI/age, while longer screen time was negatively associated with complete meal practices and physical activity. In the afternoon shift, a greater tendency towards an evening chronotype among adolescents was associated with reduced complete meal practices and lower regular fruit and vegetable consumption. Sleep duration was positively associated with a greater likelihood of a complete meal practices and regular fruit consumption. Moreover, longer screen time was positively associated with unhealthy food consumption. In both shifts, physically active adolescents consumed fruits more regularly. The practice of regular consumption of complete meals, fruits, vegetables, or unhealthy foods, in addition to being directly influenced by chronotype and other variables, was also influenced by shift. Adolescents with shorter sleep duration were at a higher risk of being overweight,

**Data availability statement:** The database and support files, essential for understanding the database and analyses, are available for download in the OSF Home online repository via the provided link. https://osf.io/ba6g3/.

**Funding:** Cintia Chaves Curioni nºE-26/211.881/2021 Fundação de Amparo à Pesquisa do Estado do Rio de Janeiro https://www.faperj.br/ The sponsors or funders did not have any role in the study.

**Competing interests:** The authors have declared that no competing interests exist.

suggesting a direct influence of sleep on anthropometric measurements. Our findings underscore the importance of considering study shifts in future interventions.

## Introduction

Adolescence, the period between 10 and 19 years of age, is characterized by profound biological and social changes. These transformations, which are increasingly influenced by the growing use of technology, have significantly affected the development of individuals, making it more challenging to develop appropriate public health policies for this age group [1–4]. To improve these policies and prevent chronic non-communicable diseases in adulthood [1,5,6] it is essential to unify and analyze study models that simultaneously consider the different health variables, technologies, and social obligations in which adolescents are included.

Recently, early health problems, such as changes in nutritional status, have become increasingly common among adolescents worldwide [7–10]. In addition to dietary habits and physical activity, sleep duration, chronotype, social jet lag, and screen time also appear to influence nutritional status, possibly through complex metabolic interactions [11–14].

Social jetlag, defined as the discrepancy between biological and social rhythms, results from differences between sleep schedules during the week and weekends. This discrepancy can affect the biological clock and alter the sleep/wake cycle, cognitive ability, and daily performance of adolescents [15,16]. The chronotype, which refers to an individual's natural predisposition to have energy peaks at certain times of the day, is an important factor to consider when evaluating biological rhythms in relation to social rhythms. People with a morning chronotype are more active in the morning, whereas those with an evening chronotype are more active in the afternoon or evening. The intermediate chronotype indicates greater adaptability to social obligations, allowing for earlier or later awakening without compromising daily performance [17,18].

Another factor that can negatively influence the sleep cycle is increased time spent on screens, such as computers, televisions, video games, or smartphones [19,20]. Exposure of adolescents to these devices before bedtime can alter their biological clock, causing a delay in the sleep phase and reducing melatonin secretion [20,21]. Social obligations, such as school, can also affect adolescents' sleep differently depending on the study shift (morning or afternoon) [22,23].

In addition to these biological factors, food practices and physical activity are influenced by chronotype, social jet lag, and screen time. High consumption of soft drinks, low consumption of fruits and vegetables, and low frequency of physical activity, in addition to an increased risk of obesity [24–29], are associated with the evening chronotype, which prefers to sleep and wake up later [15,27,28]. High social jet lag and increased daily screen time have also been associated with poor food practices [13,14,27,30].

Although there are specific studies on the associations among chronotype, sleep duration, social jet lag, screen time, physical activity, and food practices, only few

have explored the integrated interrelationships among these factors in adolescents attending different school shifts. The lack of such a comprehensive analysis limits a deeper understanding of how lifestyle and chronobiological predispositions influence adolescent health.

Therefore, this study aimed to explore the interrelationships among chronobiological factors, health practices, and anthropometry in adolescents attending morning and afternoon shift classes.

## Materials and methods

### Study design and sample

This cross-sectional study was conducted at a federal educational institution in Rio de Janeiro from October 2022 to July 2023, with adolescents enrolled in morning (7:00–12:00) and afternoon (13:00–18:00) school shifts. Anthropometric measurements of weight and height were collected in addition to completing an online questionnaire on the Google Forms digital self-completion platform. Participation in the study was voluntary, and students had the option of not answering any questions in the questionnaire if they did not feel comfortable. Pregnant and lactating adolescents and those aged more than 19 years were excluded. Of the 1,403 enrolled students, 969 agreed to participate; after applying the exclusion criteria, 925 were included in the final sample (Fig 1).

Anthropometric measurements and an online questionnaire were used to test a previously defined theory. Initially, we structured a complex theoretical model capable of assessing the interrelationships among sleep duration on school days, chronotype, social jet lag, total screen time, screen time before bedtime, physical activity level, food practices, and nutritional status, controlling for socioeconomic characteristics and stratifying by school shift. Fig 2 illustrates the assumptions regarding causal relationships between the variables and potential confounding variables (Fig 2).

### Food practices

Initially, two latent variables (constructs) were proposed to represent healthy and unhealthy food practices. The latent variables were designed based on items that assessed weekly food consumption categorized as healthy (beans, vegetables, and fruits) or unhealthy (fried snacks, sweet treats, soda, processed snacks, and fast food) [31]. We also considered the regular consumption of food five times or more during the week of the three main daily meals that constitute the normal eating routines – breakfast, lunch, and dinner – which, according to the Food Guide for the Brazilian population, provide approximately 90% of the daily caloric intake [32]. Regular consumption of the latent variables was defined as five times

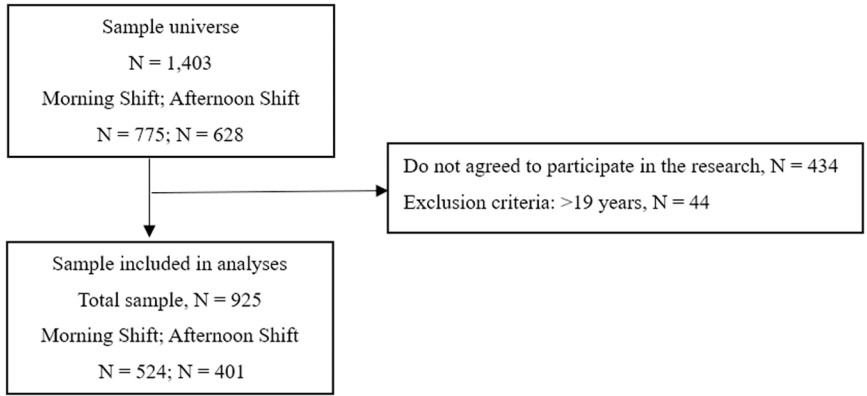

**Fig 1. Study participation flowchart.**

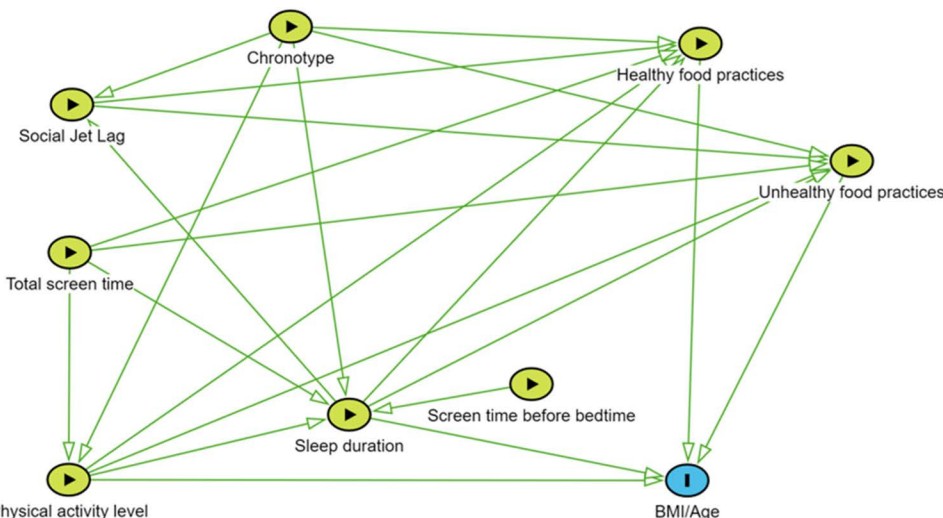

**Fig 2. Schematic representation of the study hypotheses.** Stratification variable: Study shift. Control variables: Age, sex, race/color, and income status.

or more a week in both analyses, except for the consumption of fast food which was considered regular when consumed three or more times a week [33].

To validate the two theoretical latent variables (healthy or unhealthy food practices), the items were subjected to an exploratory factor analysis [34], which resulted in grouping into three distinct factors, all with factor loadings of 0.3 or more: Factor 1: breakfast, lunch, dinner, and beans; Factor 2: fried snacks, sweet treats, soda, processed snacks, and fast food; and Factor 3: vegetables and fruits.

Consequently, the results were confirmed through a confirmatory factor analysis in which all items also presented a factor loading of 0.3 or more. However, to form latent variables, it is necessary to have at least three variables in the same group [35]. As the items "vegetables" and "fruits" were allocated to the same factor but consisted of only two variables, they were analyzed individually and independently in the structural equation model, without forming a new latent variable. Given the conceptual recognition of items associated with healthy eating within Factors 1 and 3 in the exploratory factor analysis, an alternative was to force the vegetable and fruit items into Factor 1, along with breakfast, lunch, dinner, and beans. However, this approach did not prove to be relevant, as these items did not exhibit satisfactory factor loadings (≥ 0.3) in either school shift according to the confirmatory factor analysis. Consequently, these items explained very little of the factor's variance due to their low correlation with the items in Factor 1, suggesting that they did not adequately represent the theoretical construct that Factor 1 intended to measure. A possible explanation for this finding is the context of the study, where the federal educational institution provides free, high-quality meals daily, reducing the variability typically seen in fruit and vegetable consumption. Additionally, these foods may exhibit greater consumption variability because they are often included in different contexts, such as snacks [32], rather than main meals, which are represented in Factor 1. This distribution contributed to their allocation within Factor 3.

As "vegetables" and "fruits" did not load in the same latent group as beans and other healthy eating routines, Factor 1 was called "complete meals," always considering them to be healthy, instead of "healthy food practices," as initially predicted. Factor 2 was called "unhealthy foods." Hence, six path models were developed that considered the variables related to eating separately (i.e., "complete meals," "unhealthy foods," "vegetables and fruits"), stratified by shift (morning or afternoon).

## Weekday sleep duration, chronotype, and social jet lag

Sleep variables were obtained using the Munich Chronotype Questionnaire (MCTQ). School-day sleep duration (SDW) was recorded in minutes and calculated as the difference between wake-up time and bedtime, including the latency period (the time it takes an adolescent to fall asleep after lying down) [36].

The chronotype was determined based on the midpoint of sleep on free days adjusted for sleep hours during school days (MSFsc) [36–38]. We could not calculate the chronotype of adolescents who needed an alarm clock to wake up on weekends and those who used sleeping pills three or more times a week. To quantify the chronotype, two formulae were used: if the sleep duration on free days was less than or equal to the sleep duration on school days, the chronotype was calculated using Formula (1); otherwise, Formula (2) was used.

$$\text{If } SDF \le SDW: \ MSFsc = MSF = SOf + \frac{SDF}{2} \tag{1}$$

$$\text{If } SDF > SDW: \ MSFsc = MSF - \frac{(SDF - SDweek)}{2} \tag{2}$$

where MSF is the midpoint of sleep on free days, SDF is the sleep duration on free days, SDW is the sleep duration on school days, SDweek is the average weekly sleep duration, and SOf is sleep onset on free days.

After calculating the chronotype in its continuous form, variations were observed, indicating the distribution of individuals along the chronobiological spectrum, where lower values indicated a tendency towards a morning chronotype and higher values indicated a tendency towards an evening chronotype [39].

To calculate the social jet lag (SJL), the value attributed to mid-sleep time on free days (MSF) and school days (MSW) was obtained [36], and then the MSF value was subtracted from the MSW, according to the formula:

$$MSF = SO + \frac{SD}{2}$$

$$MSW = SO + \frac{SD}{2}$$

$$SJL = MSF - MSW$$

where SJL is social jet lag; MSF is mid-sleep on free days; MSW is mid-sleep on school days; SO is sleep onset (time); and SD is sleep duration (h).

## Physical activity level

Physical activity levels were assessed using the reduced version of the International Physical Activity Questionnaire validated for the Brazilian population [40]. Adolescents who accumulated at least 300 min of physical activity per week, ranging from moderate to vigorous intensity, were classified as active [41].

## Total screen time and screen time before bedtime

Total screen time was obtained through a question adapted from the National Survey of School Health (PeNSE) questionnaire, which assesses the total daily time spent using technologies with screens on weekdays [31]. The students had nine answer options varying between "up to 1 h per day" and "more than 8 h per day." Screen time before bedtime was assessed through a question about the time spent using electronic devices before bed, with seven answer options varying from "until falling asleep" to "more than 2 h before." For both variables, the response options were categorized into nine groups for total screen time and seven for screen time before bedtime. These categories were treated as ordinal variables.

 

## Anthropometric nutritional status

Weight and height measurements were obtained using an electronic scale and a portable stadiometer (Avanutri), respectively, following standardized protocols [42] in an appropriate location previously agreed with the students. The Z-score of the body mass index by age (BMI/age) was calculated using the World Health Organization (WHO) Anthro Plus program and treated as a continuous variable in this study.

## Assessment of potential covariates

Socioeconomic status was calculated based on questions from the PenSe 2019 related to the family's economic conditions, possession of goods and services, and maternal education [33]. To create a socioeconomic score, the answers were subjected to a principal component analysis. Subsequently, the score of the most significant component within the income category was used, and the sample was divided into income quartiles. Data on school shifts, sex, age, and race/color were collected to adjust the model [43].

## Ethical standards

The Free and Informed Consent Form was distributed to the parents or guardians of the enrolled adolescents and the Free and Informed Assent Form the students, following ethical precepts, in accordance with Resolution No. 466/2012 of the National Health Council/ Ministry of Health. Both forms were delivered as signed forms. This study was approved by the Research Ethics Committee of the State University of Rio de Janeiro (approval number: 45812221.7.0000.5282).

## Statistical analysis

To analyze the interrelationships between all the variables involved in the study, structural equation modeling was applied using JASP software version 0.18.1. All analyses were initially adjusted for age, sex, race/color, and income score. Those with a p-value of less than 0.05 were considered to have a significant contribution to each relationship between the variables in the models. In the methodology for excluding adjustment variables, the variable with the highest p-value was removed first, and this procedure was repeated until only variables with a significant contribution to the adjustment indices of the models remained. In the analyses involving the BMI/age outcome, the adjustment variables were maintained regardless of the p-value owing to the robust association with these outcomes, according to previously established criteria [44]. Subsequently, the power of each of the six models was verified based on the recommendation that the sample size should be at least 10 times the number of free parameters of the model [45]. All the models met this recommendation.

The models were estimated using a Robust Maximum Likelihood estimator. To handle missing data, an advanced full-information maximum likelihood (FIML) technique was added to the analyses. The FIML technique does not exclude cases with missing data. Instead, it uses all the available observations for each case, even if some values are missing. This means that if a participant provided responses for some variables but not others, the FIML still used the available information from that participant to contribute to the estimation of the parameters. The fit of the measurement and path models was assessed using fit indices, including the chi-square (p-value>0.05), the ratio of chi-square to degrees of freedom ($X^2$/df<5), the comparative fit index (CFI ≥ 0.90), the Tucker–Lewis index (TLI > 0.90), the root mean square error fit index (RMSEA ≤ 0.06), and the standardized root mean square residual (SRMR < 0.08) [46]. The post-estimation command 'modification indices' was initially considered to improve the fit of the models by adding theoretically sound correlations between error terms. However, no modifications were implemented, as the structural equation models already met the recommended fit criteria without requiring additional adjustments. The standardized coefficients are presented in the Results section.

## Results

Of the 925 eligible adolescents who participated in the survey, 56.6% and 43.3% were enrolled in the morning and afternoon shifts, respectively. Table 1 presents the sample characteristics for ordinal and dichotomous variables according to study shift, and Table 2 presents the continuous variables.

**Table 1. Sample characterization by school shift.**

| | N (%) | Morning (%) (n = 524) | Afternoon (%) (n = 401) | P-Value |
|---|---|---|---|---|
| *Individual characteristics* | | | | |
| *Race/Color (total 903\*\*)* | | | | |
| White | 460 (50.9) | 273 (53.2) | 187 (48) | 0.117 |
| Non-whites | 443 (49.1) | 240 (46.8) | 203 (52) | – |
| *Sex (total 925)* | | | | |
| Male | 530 (57.3) | 289 (55.2) | 241 (60.1) | 0.132 |
| Female | 395 (42.7) | 235 (44.8) | 160 (39.9) | – |
| *Eating Routines - Weekly Frequency* | | | | |
| *Breakfast (total 925)* | | | | |
| < 5 days/ week | 383 (41.4) | 197 (37.6) | 186 (46.4) | 0.007\* |
| ≥ 5 days/ week | 542 (58.6) | 327 (62.4) | 215 (53.6) | – |
| *Lunch ᵃ (total 925)* | | | | |
| < 5 days/ week | 121 (13.1) | 50 (9.5) | 71 (17.7) | <0.001\* |
| ≥ 5 days/ week | 804 (86.9) | 474 (90.5) | 330 (82.3) | – |
| *Dinner (total 925)* | | | | |
| < 5 days/ week | 237 (25.6) | 157 (30) | 80 (20) | <0.001\* |
| ≥ 5 days/ week | 688 (74.4) | 367 (70) | 321 (80) | – |
| *Weekly consumption of healthy and unhealthy eating marker foods* | | | | |
| *Bean (total 925)* | | | | |
| < 5 days/ week | 361 (39) | 196 (37.4) | 165 (41.1) | 0.248 |
| ≥ 5 days/ week | 564 (61) | 328 (62.6) | 236 (58.9) | – |
| *Vegetables ᵇ (total 925)* | | | | |
| < 5 days/ week | 533 (57.6) | 272 (51.9) | 261 (65.1) | <0.001\* |
| ≥ 5 days/ week | 392 (42.4) | 252 (48.1) | 140 (34.9) | – |
| *Fruits ᶜ (total 925)* | | | | |
| < 5 days/ week | 656 (71) | 379 (72.3) | 277 (69.1) | 0.281 |
| ≥ 5 days/ week | 269 (29) | 145 (27.7) | 124 (30.9) | – |
| *Fried snacks ᵈ (total 925)* | | | | |
| < 5 days/ week | 861 (93.1) | 495 (94.5) | 366 (91.3) | 0.058 |
| ≥ 5 days/ week | 64 (6.9) | 29 (5.5) | 35 (8.7) | – |
| *Sweet treats (total 925)* | | | | |
| < 5 days/ week | 583 (63) | 341 (65.1) | 242 (60.3) | 0.140 |
| ≥ 5 days/ week | 345 (37) | 183 (34.9) | 159 (39.7) | – |
| *Soda (total 925)* | | | | |
| < 5 days/ week | 793 (85.7) | 451 (86.1) | 342 (85.3) | 0.736 |
| ≥ 5 days/ week | 132 (14.3) | 73 (13.9) | 59 (14.7) | – |
| *Processed snacks ᵉ (total 925)* | | | | |
| < 5 days/ week | 637 (68.9) | 365 (69.7) | 272 (67.8) | 0.552 |
| ≥ 5 days/ week | 288 (31.1) | 159 (30.3) | 129 (32.2) | – |
| *Fast Food ᶠ (total 925)* | | | | |
| < 3 days/ week | 783 (84.7) | 451 (86.1) | 332 (82.8) | 0.171 |
| ≥ 3 days/ week | 142 (15.3) | 73 (13.9) | 69 (17.2) | – |
| *Physical activity level* | | | | |
| *Physical activity level (923\*\*)* | | | | |
| Insufficiently active | 569 (61.6) | 318 (60.8) | 251 (62.7) | 0.547 |
| Sufficiently active | 354 (38.4) | 205 (39.2) | 149 (37.3) | – |

*(Continued)*

**Table 1.** (Continued)

| | N (%) | Morning (%) (n = 524) | Afternoon (%) (n = 401) | P-Value |
|---|---|---|---|---|
| *Income status* | | | | |
| *Income status in quartiles (total 882***)* | | | | |
| 1º Income quartile | 219 (24.8) | 115 (22.9) | 104 (27.4) | 0.129 |
| 2º, 3 andº 4º Income quartiles | 663 (75.2) | 387 (77.1) | 276 (72.6) | – |
| *Screen time* | | | | |
| *Total screen time (925)* | | | | |
| Up to 1 hour per day | 30 (3.2) | 17 (3.2) | 13 (3.2) | 0.546 |
| Between 1 hour and 2 hours per day | 68 (7.4) | 32 (6.1) | 36 (9) | – |
| Between 2 hours and 3 hours per day | 100 (10.8) | 51 (9.7) | 49 (12.2) | – |
| Between 3 hours and 4 hours per day | 126 (13.6) | 71 (13.5) | 55 (13.7) | – |
| Between 4 hours and 5 hours per day | 145 (15.7) | 82 (15.6) | 63 (15.7) | – |
| Between 5 hours and 6 hours per day | 136 (14.7) | 78 (14.9) | 58 (14.5) | – |
| Between 6 hours and 7 hours per day | 110 (11.9) | 68 (13) | 42 (10.5) | – |
| Between 7 hours and 8 hours per day | 63 (6.8) | 34 (6.5) | 29 (7.2) | – |
| More than 8 hours per day | 147 (15.9) | 91 (17.4) | 56 (14) | – |
| *Screen Time Before bedtime (925)* | | | | |
| Until falling asleep | 339 (36.7) | 185 (35.3) | 154 (38.4) | 0.321 |
| Up to 15 minutes before | 361 (35.1) | 218 (41.6) | 144 (35.9) | – |
| Up to 30 minutes before | 124 (13.4) | 72 (13.7) | 52 (13) | – |
| Up to 1 hour before | 45 (4.9) | 20 (3.8) | 25 (6.2) | – |
| Up to 1 hour and a half before | 14 (1.5) | 8 (1.5) | 6 (1.5) | – |
| Up to 2 hours before | 17 (1.8) | 7 (1.3) | 10 (2.5) | – |
| More than 2 hours before | 24 (2.6) | 14 (2.7) | 10 (2.5) | – |

* P value <0.05 **Some did not know or preferred not to respond *** Some did not know their mother's education.

The comparison between groups of categorical variables was conducted using a chi square test.

a Consider meals containing rice, beans, meat, salad, and cooked vegetables such as lunch, soup, pasta, etc., excluding sandwiches.

b Lettuce, chard, watercress, arugula, collard, cabbage, broccoli, spinach, other greens, tomato, cucumber, carrot, beetroot, pumpkin, zucchini, eggplant, okra, etc., excluding potatoes.

c Fruits or fruit salad

d French fries (not including packaged potatoes) or fried snacks such as chicken drumsticks, fried kibbeh, and fried pastries, etc.

e Hamburger, ham, mortadella, salami, sausage, instant noodles, packaged snacks, and savory biscuits

f Fast food restaurants such as cafeterias, hot dog stands, pizzerias, and hamburgers, etc.

In the morning shift, the predominant characteristics were as follows: white adolescents (53.2%), males (55.2%), insufficient physical activity levels (60.8%), total daily screen use for around 4–5 h (15.6%), and screen use up to 15 min before bedtime (41.6%). In the afternoon shift, the predominant characteristics were: white adolescents (48%), males (60.1%), insufficient physical activity levels (62.7%), total daily screen use for around 4–5 h (15.7%), and screen use up to 15 min before bedtime (38.4%) (Table 1).

When evaluating the variables in Table 2, we found that students enrolled in the morning shift tended to have lower chronotype scores, indicating an inclination toward morningness, whereas those who studied in the afternoon tended to have higher chronotype scores, indicating an inclination toward eveningness (p < 0.001). Furthermore, students enrolled in the morning shift had shorter sleep durations on school days (p < 0.001) and greater social jet lag (p < 0.001) than those enrolled in the afternoon shift. There was no significant difference in nutritional status between students studying in the different shifts (p = 0.417).

**Table 2. Characterization of continuous variables by study shift.**

| Continuous variables | N | % | Normality test p-value | Mean or median | SD or IQR | P-Value |
|---|---|---|---|---|---|---|
| *Chronotype** *Morning* *Afternoon* *Sleep duration on school days (minutes)* | 420 314 | 51.2 42.8 | 0.008 0.226# | 3.79 5.04 | 1.85 1.62 | <0.001 |
| *Morning* *Afternoon* | 524 401 | 56.7 43.3 | 0.001 0.002 | 360 470 | 97,5 120 | <0.001 |
| *Social Jet lag (minutes)* | | | | | | |
| *Morning* *Afternoon* | 524 401 | 56.7 43.3 | 0.001 0.001 | 155 90 | 120 105 | <0.001 |
| *Age (years)* | | | | | | |
| *Morning* *Afternoon* | 524 401 | 56.7 43.3 | 0.001 0,001 | 16.5 16 | 3 3 | 0.131 |
| *BMI/Age (score-z)*** | | | | | | |
| *Morning* *Afternoon* | 502 383 | 56.7 43.3 | 0.057# 0.294# | 0.24 0.17 | 1.26 1.35 | 0.417 |

SD: Standard deviation IQR: Interquartile range

\* Participants who needed an alarm clock to wake up on weekends were excluded.

\*\* Some preferred not to take the measurements

\# Variables with normal distribution according to the Shapiro-Wilk test were expressed as mean and standard deviation. Variables with non-normal distribution were expressed as median and interquartile range.

Note: The comparison between the morning and afternoon shifts was performed using the independent samples t-test (also known as the two-sample t-test) when both groups showed a normal distribution. However, if one or both groups showed a non-normal distribution for the same variable, the Mann-Whitney test was used.

## Path models

Based on our methodology, six path models were developed to evaluate the variables related to nutrition, divided into "complete meals," "unhealthy foods," or "vegetables and fruits," with stratification by school shift (morning or afternoon). All the variables listed in Tables 1 and 2 were included in these models (Figs 3 and 4), which mostly met the adjustment indices recommended in the literature (S1 Table). All models showed fit indices within recommended criteria (CFI ≥ 0.90; TLI > 0.90; RMSEA ≤ 0.06; SRMR < 0.08), although slight variations were observed. For instance, CFI values ranged from 0.933 to 0.963, and TLI values ranged from 0.906 to 0.955, with slightly lower values in Models 5 and 6. RMSEA values also varied but consistently remained within an acceptable range. Although the χ² test was significant for most models - likely due to the large sample size - the χ²/df ratio remained acceptable across all models (< 5), supporting the overall robustness of the model fits [45,46]. Table 3 summarizes the direct effects of these variables.

### Morning shift models – direct effects

Table 3 and Fig 3 show that a greater tendency towards an evening chronotype among adolescents was associated with an increase in social jet lag and shorter sleep duration on school days. Reduced sleep duration was associated with an increase in BMI/age.

Although no direct association was found between food practices or physical activity levels and BMI/age, the models for the morning shift indicated that a higher probability of regular fruit and vegetable consumption was associated with students who were more physically active. Furthermore, both physical activity levels and complete meal consumption were negatively associated with total screen time.

**Fig 3. Structural equation modeling for students enrolled in the morning shift. Models 1, 3 and 5.** *p ≤ 0.05. # Factor loading of the item. Schedules: Morning classes start at 7:00 AM/ Morning classes end at 12:00 PM.

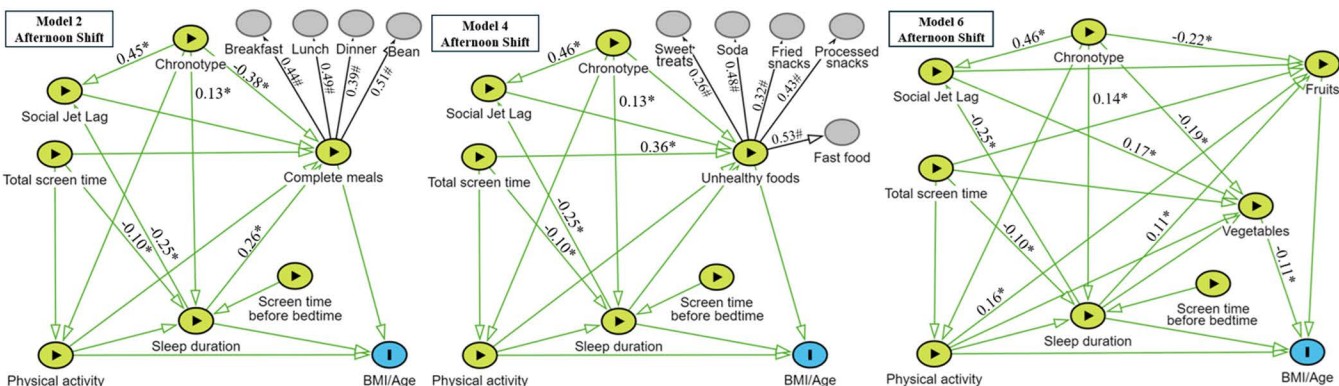

**Fig 4. Structural equation modeling for students enrolled in the afternoon shift. Models 2, 4 and 6.** *p ≤ 0.05. # Factor loading of the item. Schedules: Afternoon classes start at 1:00 PM/ Afternoon classes end at 6:00 PM.

## Afternoon-shift models – direct effects

In the afternoon-shift models (Table 3 and Fig 4), a greater tendency towards an evening chronotype was associated with an increase in social jet lag, longer sleep duration on school days, lower adherence to complete meal practices, and lower regular fruit and vegetable consumption. Furthermore, regular vegetable consumption was negatively associated with BMI/age.

Although no direct associations were observed among complete meal practices, unhealthy foods, regular fruit consumption, or physical activity levels and BMI/age, the afternoon shift models suggested that students who were more physically active and had longer sleep durations were more likely to consume fruit regularly. Sleep duration was also positively associated with a greater likelihood of eating complete meals. In contrast, high daily screen time was associated with a greater chance of regular consumption of unhealthy foods and reduced sleep duration.

## Afternoon-shift models - indirect and total effects

To understand the unexpected combinations of relationships observed - a positive association between chronotype and sleep duration in the afternoon shift, a positive relationship between sleep duration and a higher likelihood of complete

**Table 3. Direct effects in models 1 to 6, stratified by school shift.**

| MODELS | | | | | | | | | | | | |
|---|---|---|---|---|---|---|---|---|---|---|---|---|
| | Model 1 Morning | | Model 2 Afternoon | | Model 3 Morning | | Model 4 Afternoon | | Model 5 Morning | | Model 6 Afternoon | |
| **Direct relations** | β | p | B | P | B | p | β | P | B | P | B | P |
| SDW→BMI/age | -0.11 | 0.01* | -0.05 | 0.31 | -0.11 | 0.01* | -0.08 | 0.16 | -0.12 | 0.01* | -0.08 | 0.16 |
| PA→BMI/age | 0.04 | 0.33 | 0.03 | 0.55 | 0.04 | 0.34 | 0.03 | 0.49 | 0.02 | 0.64 | 0.04 | 0.49 |
| CM→BMI/age | 0.03 | 0.67 | -0.10 | 0.21 | – | – | – | – | – | – | – | – |
| UHF→BMI/age | – | – | – | – | -0.03 | 0.58 | 0.02 | 0.73 | – | – | – | – |
| Vegetables→BMI/age | – | – | – | – | – | – | – | – | 0.06 | 0.16 | -0.11 | 0.03* |
| Fruits→BMI/age | – | – | – | – | – | – | – | – | 0.13 | 0.01* | 0.06 | 0.23 |
| SDW→CM | -0.01 | 0.88 | 0.26 | 0.00* | – | – | – | – | – | – | – | – |
| SDW→UHF | – | – | – | – | -0.01 | 0.90 | -0.04 | 0.54 | – | – | – | – |
| SDW→Vegetables | – | – | – | – | – | – | – | – | 0.04 | 0.43 | 0.07 | 0.19 |
| SDW→Fruits | – | – | – | – | – | – | – | – | 0.04 | 0.32 | 0.11 | 0.02* |
| Chronotype→CM | -0.05 | 0.67 | -0.38 | 0.00* | – | – | – | – | – | – | – | – |
| Chronotype→UHF | – | – | – | – | 0.19 | 0.09 | -0.03 | 0.77 | – | – | – | – |
| Chronotype→Vegetables | – | – | – | – | – | – | – | – | 0.03 | 0.72 | -0.19 | 0.00* |
| Chronotype→ Fruits | – | – | – | – | – | – | – | – | 0.08 | 0.30 | -0.22 | 0.00* |
| SJL→CM | 0.06 | 0.59 | 0.08 | 0.35 | – | – | – | – | – | – | – | – |
| SJL→UHF | – | – | – | – | -0.07 | 0.51 | 0.04 | 0.64 | – | – | – | – |
| SJL →Vegetables | – | – | – | – | – | – | – | – | -0.06 | 0.42 | 0.17 | 0.00* |
| SJL→ Fruits | – | – | – | – | – | – | – | – | -0.05 | 0.50 | 0.10 | 0.07 |
| TST→CM | -0.17 | 0.01* | 0.02 | 0.77 | – | – | – | – | – | – | – | – |
| TST→UHF | – | – | – | – | 0.12 | 0.10 | 0.36 | 0.00* | – | – | – | – |
| TST→Vegetables | – | – | – | – | – | – | – | – | -0.08 | 0.09 | -0.05 | 0.30 |
| TST→ Fruits | – | – | – | – | – | – | – | – | -0.07 | 0.11 | 0.03 | 0.52 |
| PA→CM | 0.06 | 0.31 | -0.07 | 0.28 | – | – | – | – | – | – | – | – |
| PA→UHF | – | – | – | – | -0.04 | 0.52 | -0.00 | 0.97 | – | – | – | – |
| PA→Vegetables | – | – | – | – | – | – | – | – | 0.11 | 0.01* | 0.09 | 0.06 |
| PA→ Fruits | – | – | – | – | – | – | – | – | 0.15 | 0.00* | 0.16 | 0.00* |
| PA→SDW | 0.00 | 0.93 | 0.02 | 0.67 | 0.00 | 0.93 | 0.02 | 0.67 | -0.00 | 0.93 | 0.02 | 0.66 |
| Chronotype→SDW | -0.19 | 0.00* | 0.13 | 0.01* | -0.18 | 0.00* | 0.13 | 0.01* | -0.19 | 0.00* | 0.14 | 0.01* |
| STB→SDW | 0.04 | 0.40 | -0.04 | 0.48 | 0.04 | 0.40 | -0.04 | 0.47 | 0.04 | 0.40 | -0.04 | 0.46 |
| TST→SDW | -0.03 | 0.51 | -0.10 | 0.05* | -0.03 | 0.50 | -0.10 | 0.05* | -0.03 | 0.51 | -0.10 | 0.04* |
| SDW→SJL | -0.01 | 0.79 | -0.25 | 0.00* | -0.01 | 0.68 | -0.25 | 0.00* | -0.01 | 0.77 | -0.25 | 0.00* |
| Chronotype→SJL | 0.83 | 0.00* | 0.45 | 0.00* | 0.83 | 0.00* | 0.46 | 0.00* | 0.83 | 0.00* | 0.46 | 0.00* |
| Chronotype→PA | 0.00 | 0.96 | -0.09 | 0.11 | 0.00 | 0.96 | -0.09 | 0.11 | -0.00 | 0.97 | -0.09 | 0.11 |
| TST→PA | -0.13 | 0.00* | 0.00 | 0.97 | -0.13 | 0.00* | 0.00 | 0.97 | -0.13 | 0.00* | 0.00 | 0.97 |

*Significant p-value. Fit indices used - χ2: Chi-square (p-value > 0.05); χ2/df: Ratio between chi-square and degrees of freedom (< 5); CFI: Comparative fit index (≥0.90); TLI: Tucker–Lewis index (>0.90); RMSEA Root mean square error fit index (≤ 0.06); SRMR: Standardized root mean square residual (<0.08). Legends: SDW: Sleep duration on school days; BMI/age: Body mass index by age; CM: Complete meals; UHF: Unhealthy foods; SJL: Social jet lag; TST: Total screen time; STB: Screen Time Before bedtime; PA: Physical activity

meal practices and regular fruit consumption, and a negative association between chronotype and these same complete meals practices - we conducted a detailed analysis of the contribution of sleep duration on food practices, taking into account chronotype. This analysis included the assessment of direct effects (Table 3), as well as indirect and total effects (S2 and S3 Tables).

When the model considered the latent variable "complete meals," the analysis revealed a positive and significant association (B = 0.033, p = 0.045) for the indirect relationship among chronotype, sleep duration, and complete meals. The total relationship between chronotype and complete meals was negative and significant (B = -0.345, p < 0.001). These results suggest that sleep duration partially mediates the relationship between chronotype and complete meal practices. The indirect effect of chronotype on complete meal practices, mediated by sleep duration, was positive and significant. This indicates that the negative relationship between chronotype and complete meal practices can be partially explained by longer sleep duration. When we considered both the direct and indirect effects of chronotype on complete-meal practices, the total effect showed a significant negative relationship. This suggests that even after accounting for the mediating effect of sleep duration, chronotype exerts a direct negative influence on complete meal practices. These results indicate that sleep duration plays an important role in explaining complete meal practices in students with different chronotypes; however, the direct influence of chronotype remains a relevant factor.

In the case of fruit consumption, the indirect relationship among chronotype, sleep duration, and fruits was positive but not significant (B = 0.015, p = 0.075), while the total relationship between chronotype and fruits remained negative and significant (B = -0.207, p < 0.001). These results suggest that sleep duration does not play a significant role in mediating chronotype and regular fruit consumption. The total effect of chronotype on regular fruit consumption, which encompassed both direct and indirect effects, remained negative and significant, although it was smaller than the direct effect alone (B = -0.224, p < 0.001, Table 3). This finding suggests that other factors may influence the relationship between chronotype and regular fruit consumption. However, even after accounting for sleep duration, the negative relationship between chronotype (greater tendency towards an evening chronotype) and regular fruit consumption remained significant.

## Discussion

This study aimed to advance the understanding of the interrelationships among chronotype, sleep duration, social jet lag, physical activity, screen time, food practices, and nutritional status in adolescents. The overarching hypothesis was that these variables were interconnected and influenced adolescents differently depending on their study shifts (morning or afternoon). This study is essential because adolescence is a critical period for establishing healthy habits that persist into adulthood. Understanding the interactions between these variables may help formulate effective interventions that promote adolescent health and well-being.

By organizing food items into latent variables, such as "complete meals" and "unhealthy foods," and by treating vegetables and fruits as direct variables, this study reveals the importance of considering study shifts when analyzing these behaviors. Our main finding was that chronotype directly influenced food practices but only among students enrolled in the afternoon shift, when there was no requirement to wake up early. There was a significant association between regular vegetable consumption and BMI/age in this group. In contrast, sleep duration was a more relevant factor for students enrolled in the morning shift, highlighting the influence of the need to wake up earlier.

Furthermore, additional findings from this study indicated that sleep duration, social jetlag, physical activity level, and total screen time also play important roles in adolescents' eating habits and nutritional status. Among all the variables studied, regular vegetable consumption was the only one that showed the expected association with BMI/age, especially in the afternoon shift, suggesting that healthy food practices may be highly effective in maintaining a healthy weight [6]. However, recent changes in these practices may not be immediately reflected in BMI/age [47,48], as in the case of morning-shift students.

An interesting finding was the negative relationship between sleep duration on school days and BMI/age, which was observed only in the morning shift. We believe that sleep deprivation caused by the need to wake up early may lead to hormonal responses that dysregulate appetite and affect hunger and satiety. These include low levels of leptin and high levels of ghrelin, which would increase appetite and contribute to increased BMI/age [49]. Changes in factors that affect metabolism, including insulin and glucose metabolism, cortisol, growth hormone, and thyroid-stimulating hormone, are also important [50]. The activation of inflammatory pathways due to short sleep may be implicated in the development of obesity and may positively or negatively regulate the expression of genes involved in oxidative stress and metabolism [51]. This study also found beneficial associations between sleep duration and complete meal practices during the afternoon shift, highlighting the importance of adequate sleep in adopting a healthier diet [52]. Furthermore, it is important to highlight the inverse or bidirectional possibility of this association as adopting a healthier diet can also improve sleep time and quality [53].

Another significant factor was total screen time, which showed a negative association with sleep duration in the afternoon shift. The longer sleep duration among afternoon students may explain why this relationship was not observed in the morning shift, a finding also reported in other studies [54,55]in which the obligation to wake up early seemed to have a stronger impact on sleep reduction.

Total screen time is considered an important marker of sedentary behavior in adolescents by the WHO [41]. This study found a negative association between total screen time and the level of physical activity among morning shift students, indicating that the greater the screen time, the lower the likelihood of adolescents being physically active. Through a systematic review with the aim of investigating the effects of screen time on the health of children and adolescents, Priftis and Panagiotakos (2023) pointed out several causes that justify this association, suggesting that increased screen time since childhood can negatively impact motor development, language, social, and interpersonal skills, in addition to reducing the frequency of play, which can impair the propensity for physical activity in later stages of life [14].

Notably, in the afternoon shift, total screen time was positively associated with the consumption of unhealthy foods, whereas in the morning shift, it was negatively associated with complete meal practices. This may be related to the fact that screen time can lead to distracted eating behavior, greater exposure to advertisements for unhealthy foods, and a preference for substituting healthier foods owing to laziness in planning or preparing a meal [13,14].

This study found positive associations between the level of physical activity and fruit consumption in both shifts, in addition to a positive relationship with vegetable consumption in the morning shift. Adolescents who engaged in regular physical activity tended to adopt better health habits, including better food practices. A higher level of physical activity appears to be associated with increased self-determined dietary regulation [56,57]. Thus, students who are physically active, according to the WHO recommendations, may also have healthier food practices, such as the higher consumption of fruits and vegetables observed in this study.

Chronotypes also showed significant differences between study shifts. In the afternoon shift, the more evening-oriented the students' chronotype, the lower their odds of engaging in complete meal practices and regular consumption of fruit and vegetables, as observed in other studies [12,28,58]. However, this association appears to depend on the absence of social timing influences, such as strict school schedules, which may explain the lack of this association in the morning shift.

In addition to the impact of chronotype on eating habits, we found an association between chronotype and social jet lag. The positive relationship between social jet lag and chronotype is well documented, regardless of the study shift. The more evening-oriented the students' chronotype, the higher the likelihood of experiencing greater social jet lag, as their biological preferences are misaligned with the schedules imposed by social demands [59,60].

A negative relationship between social jet lag and sleep duration on school days, observed only among afternoon-shift students, was expected. Students attending school in the afternoon tend to maintain similar sleep habits on weekdays as those on free days, resulting in longer sleep duration and, consequently, lower social jet lag than their morning shift counterparts [59].

## Inconclusive hypotheses

The positive association between regular fruit consumption and BMI/age, observed only in the morning shift, may seem counterintuitive. Furthermore, no association was found between the physical activity level and BMI/age, ruling out the effect of regular physical activity. However, we did not have access to data on daily fruit intake, which makes it difficult to suggest a direct causal relationship. Furthermore, the National School Feeding Program, a public policy adopted in Brazil that offers daily fruit to students, may have contributed to the increased consumption [61], but without enough time to generate a significant reduction in BMI. This should be considered a positive factor, as it promotes healthy food practices, although its effects on BMI may be more visible in the long term [47,48]. The same is true for physical activity. The body may take time to respond to changes in diet and physical activity levels [62]. In the short term, BMI/age may not decrease even if a person adopts healthier combined behaviors, such as those observed in relation to physical activity and fruit consumption.

The positive association between social jet lag and vegetable consumption may be linked to the daily provision of healthy foods in schools, which contradicts the literature that generally associates social jet lag with poor food practices [27,30].

Regarding sleep duration, there was a negative association with chronotype among morning-shift students and a positive association among afternoon-shift students. Morning-shift students with an evening chronotype tend to have shorter sleep durations as they prefer to wake up later [28]. In the afternoon shift, no significant association was expected because, theoretically, everyone would follow their preferred sleep routine. However, this positive association could be explained by the temporary need to wake up earlier for extra activity in the morning, which may lead to sleep compensation on other days.

We did not find a significant association between screen time before bedtime and total sleep duration. However, the negative relationship observed between total daily screen time and sleep duration on school days during the afternoon shift suggests that excessive exposure to screens during the day may affect sleep, which remains an important aspect to be considered in interventions aimed at improving adolescent health.

Finally, the expected negative association between chronotype and physical activity, indicating that adolescents who are more inclined towards the evening chronotype would be less physically active, was not observed. As we did not have access to the timing of the exercise sessions, it was not possible to provide further explanations for the absence of this finding. Despite the well-established positive associations between physical activity and various health parameters, including sleep quality and duration [63,64], this study found no association between physical activity and sleep duration.

## Strengths and limitations

One of the strengths of this study was the inclusion of individuals between the ages of 14 and 19 years, allowing for the collection of a variety of information during important physical, emotional, and behavioral changes. This made the results more representative and applicable to the general adolescent population, thereby increasing the external validity of the study. Furthermore, the homogeneity of the population reduced the variability between participants, facilitating the identification of associations of interest. The diversity of the collected data allowed the exploration of different and complementary information. Furthermore, the sample size, suitable for statistical modeling with an availability of between 10 and 20 individuals per estimated parameter, strengthened the precision and reliability of the analyses and ensured stable estimates [45].

However, this study also had a few limitations. First, owing to its cross-sectional design, we were unable to infer causal relationships. Although this study suggests possible causal associations, the directions of these relationships should be confirmed through longitudinal studies. Since exposure and outcome are measured simultaneously in cross-sectional studies, it remains unclear whether the observed relationships are unidirectional or bidirectional. Thus, despite identifying the relationships between variables and discussing their possible biological mechanisms, it may not be possible to integrate all these variables into a single, coherent, and targeted biological mechanism.

Second, the study did not account for potential participation in extracurricular morning activities among students enrolled in the afternoon shift, which may have influenced certain variables. Furthermore, multiple biological mechanisms may be operating concurrently in this population, along with unmeasured confounders. Despite statistical adjustments, residual confounding cannot be entirely ruled out. While our findings align with existing literature, future longitudinal or experimental studies are needed to confirm these relationships and explore their underlying mechanisms. Finally, the use of self-reported data, although widely used in epidemiological research, is subject to biases that can compromise the accuracy of the information. In the case of food practices, students may underreport or overreport food consumption due to recall bias or social desirability bias. In addition, especially in this case, although the use of a questionnaire that explores food practices through markers of healthy and unhealthy eating allows for a larger number of participants, quantitative details of macronutrients and micronutrients, which could help explain the relationship between sleep and food, were not covered. Similarly, the estimation of screen time and sleep duration is subject to individual perception, leading to potential discrepancies. For physical activity, self-report questionnaires may not accurately reflect the intensity and duration of activities performed, emphasizing the need for caution when interpreting the results. Future research should consider objective measurement tools, such as actigraphy, food diaries, or accelerometers, to enhance data accuracy.

## Conclusion

When evaluating the interrelationship among sleep indicators, social jet lag, screen time, physical activity, and food practices in relation to anthropometric measures among adolescents attending different study shifts, it was concluded that the timing of social obligations, represented by the study shift, exerted a significant influence, suggesting that it serves as a strong environmental marker. During the afternoon shift, regular vegetable consumption was associated with a reduction in BMI/age, whereas sleep duration was positively related to complete meal practices and fruit consumption. In contrast, in morning-shift students, sleep duration had a direct negative relationship with BMI/age, independent of food practices. Furthermore, the total daily screen time was negatively associated with sleep duration, positively related to the consumption of unhealthy foods in afternoon-shift students, and negatively associated with complete meal practices and regular vegetable consumption in morning-shift students. Students with an evening chronotype who were enrolled in the afternoon shift tended to exhibit negative associations with complete meal practices and regular fruit and vegetable consumption. In both shifts, adolescents who were physically active demonstrated better food practices. Although structural equation modeling in a cross-sectional study does not allow for causal inferences, our findings underscore the importance of considering study shifts in future local-level interventions. This consideration could serve as a valuable tool for more accurate analysis of the complex and interrelated variables that impact health habits.

## Supporting information

**S1 Table. Model fit indices.**
(DOCX)

**S2 Table. Indirect and total effects in models 1, 3 and 5, stratified by morning study shift.**
(DOCX)

**S3 Table. Indirect and total effects in models 2, 4 and 6, stratified by afternoon study shift.**
(DOCX)

## Acknowledgments

We would like to thank Editage (www.editage.com) for their support on the manuscript.

## Author contributions

**Conceptualization:** Raphael Corrêa Martins, Cintia Chaves Curioni.

**Formal analysis:** Raphael Corrêa Martins, Flávia dos Santos Barbosa Brito.

**Funding acquisition:** Cintia Chaves Curioni.

**Investigation:** Raphael Corrêa Martins.

**Methodology:** Raphael Corrêa Martins, Flávia dos Santos Barbosa Brito, Cintia Chaves Curioni.

**Project administration:** Cintia Chaves Curioni.

**Supervision:** Flávia dos Santos Barbosa Brito, Cintia Chaves Curioni.

**Writing – original draft:** Raphael Corrêa Martins.

**Writing – review & editing:** Raphael Corrêa Martins, Flávia dos Santos Barbosa Brito, Cintia Chaves Curioni.

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
