## [Decision Letter · Decision Letter 0]

5 Mar 2025

PONE-D-25-00055Influence of study shift on the interrelationships among chronobiological factors, health practices, and anthropometry in adolescentsPLOS ONE

Dear Dr. Curioni,

Thank you for submitting your manuscript to PLOS ONE. After careful consideration, we feel that it has merit but does not fully meet PLOS ONE’s publication criteria as it currently stands. Therefore, we invite you to submit a revised version of the manuscript that addresses the points raised during the review process. Both reviewers have provided positive feedback and have suggested minor revisions.

To ensure clarity and consistency, we kindly request that you address these minor revisions, which will further strengthen your manuscript. Please submit your revised manuscript along with a detailed response to the reviewers’ comments. Best regards,

Dr. Miray Budak

Reviewer 1

The manuscript is technically sound and the use of structural equation modeling (SEM) to analyze interrelationships is an advanced and appropriate choice for dealing with multiple variables and testing theoretical frameworks.

Regarding English: in Table 2 there is text that is written in portuguese language. Please develop the limitation section including the reliance on self-reported data (e.g., food practices, screen time, and sleep duration) that may introduce bias or inaccuracies.

Reviewer 2

I have reviewed your manuscript titled “Influence of study shift on the interrelationships among chronobiological factors, health practices, and anthropometry in adolescents.” The study investigates how school shift—morning versus afternoon—affects the complex interrelationships among sleep-related variables, chronotype, screen time, physical activity, dietary practices, and anthropometric measures in adolescents. Using a cross-sectional design with a sample of 925 students aged 14 to 19, you collected data through an online questionnaire and anthropometric measurements. Structural equation modeling was applied to test six path models that explore direct, indirect, and total effects among the variables, with particular attention given to differences between the morning and afternoon shifts. Your findings highlight that factors such as sleep duration, chronotype, and screen time have distinct associations with food practices and body mass index by age, depending on the study shift. These results suggest that the timing of school attendance is an important environmental marker that should be considered in health interventions targeting adolescents.

While your work is comprehensive and methodologically detailed, there are several aspects that could be strengthened. First, the cross-sectional design limits the ability to infer causality, which you acknowledge; however, further discussion on this limitation and its implications for interpreting the results would improve the manuscript. Second, the reliance on self-reported measures for variables such as sleep duration, food practices, and screen time introduces potential bias. It might be useful to discuss how these biases could affect your findings or consider supplementing the questionnaire data with objective measures in future studies. Third, although the formation of latent variables for food practices is innovative, the rationale behind the grouping of certain items—particularly the separate treatment of vegetables and fruits—could be explained more thoroughly to clarify why these items did not load with other healthy eating indicators. Additionally, while the structural equation models appear robust, a more detailed presentation of the model fit indices and any modifications made based on the “modification indices” would enhance transparency.

In summary, your study provides valuable insights into the influence of school shift on adolescent health behaviors and nutritional status. Addressing these methodological and interpretative issues would further solidify the conclusions and offer a clearer pathway for future interventions.

We look forward to receiving your revised manuscript.

Kind regards,

Miray Budak

Academic Editor

PLOS ONE

Journal Requirements:

3. We notice that your supplementary tables are included in the manuscript file. Please remove them and upload them with the file type 'Supporting Information'. Please ensure that each Supporting Information file has a legend listed in the manuscript after the references list.

Reviewers' comments:

Reviewer 1

The manuscript is technically sound and the use of structural equation modeling (SEM) to analyze interrelationships is an advanced and appropriate choice for dealing with multiple variables and testing theoretical frameworks.

Regarding English: in Table 2 there is text that is written in portuguese language. Please develop the limitation section including the reliance on self-reported data (e.g., food practices, screen time, and sleep duration) that may introduce bias or inaccuracies.

Reviewer 2

I have reviewed your manuscript titled “Influence of study shift on the interrelationships among chronobiological factors, health practices, and anthropometry in adolescents.” The study investigates how school shift—morning versus afternoon—affects the complex interrelationships among sleep-related variables, chronotype, screen time, physical activity, dietary practices, and anthropometric measures in adolescents. Using a cross-sectional design with a sample of 925 students aged 14 to 19, you collected data through an online questionnaire and anthropometric measurements. Structural equation modeling was applied to test six path models that explore direct, indirect, and total effects among the variables, with particular attention given to differences between the morning and afternoon shifts. Your findings highlight that factors such as sleep duration, chronotype, and screen time have distinct associations with food practices and body mass index by age, depending on the study shift. These results suggest that the timing of school attendance is an important environmental marker that should be considered in health interventions targeting adolescents.

While your work is comprehensive and methodologically detailed, there are several aspects that could be strengthened. First, the cross-sectional design limits the ability to infer causality, which you acknowledge; however, further discussion on this limitation and its implications for interpreting the results would improve the manuscript. Second, the reliance on self-reported measures for variables such as sleep duration, food practices, and screen time introduces potential bias. It might be useful to discuss how these biases could affect your findings or consider supplementing the questionnaire data with objective measures in future studies. Third, although the formation of latent variables for food practices is innovative, the rationale behind the grouping of certain items—particularly the separate treatment of vegetables and fruits—could be explained more thoroughly to clarify why these items did not load with other healthy eating indicators. Additionally, while the structural equation models appear robust, a more detailed presentation of the model fit indices and any modifications made based on the “modification indices” would enhance transparency.

In summary, your study provides valuable insights into the influence of school shift on adolescent health behaviors and nutritional status. Addressing these methodological and interpretative issues would further solidify the conclusions and offer a clearer pathway for future interventions.

Reviewer's Responses to Questions

**Comments to the Author**

1. Is the manuscript technically sound, and do the data support the conclusions?

Reviewer #1: Yes

Reviewer #2: Yes

2. Has the statistical analysis been performed appropriately and rigorously? 

Reviewer #1: Yes

Reviewer #2: Yes

3. Have the authors made all data underlying the findings in their manuscript fully available?

Reviewer #1: Yes

Reviewer #2: Yes

4. Is the manuscript presented in an intelligible fashion and written in standard English?

Reviewer #1: Yes

Reviewer #2: Yes

5. Review Comments to the Author

Reviewer #1: The manuscript is technically sound and the use of structural equation modeling (SEM) to analyze interrelationships is an advanced and appropriate choice for dealing with multiple variables and testing theoretical frameworks.

Regarding English: in Table 2 there is text that is written in portuguese language. Please develop the limitation section including the reliance on self-reported data (e.g., food practices, screen time, and sleep duration) that may introduce bias or inaccuracies.

Reviewer #2: I have reviewed your manuscript titled “Influence of study shift on the interrelationships among chronobiological factors, health practices, and anthropometry in adolescents.” The study investigates how school shift—morning versus afternoon—affects the complex interrelationships among sleep-related variables, chronotype, screen time, physical activity, dietary practices, and anthropometric measures in adolescents. Using a cross-sectional design with a sample of 925 students aged 14 to 19, you collected data through an online questionnaire and anthropometric measurements. Structural equation modeling was applied to test six path models that explore direct, indirect, and total effects among the variables, with particular attention given to differences between the morning and afternoon shifts. Your findings highlight that factors such as sleep duration, chronotype, and screen time have distinct associations with food practices and body mass index by age, depending on the study shift. These results suggest that the timing of school attendance is an important environmental marker that should be considered in health interventions targeting adolescents.

While your work is comprehensive and methodologically detailed, there are several aspects that could be strengthened. First, the cross-sectional design limits the ability to infer causality, which you acknowledge; however, further discussion on this limitation and its implications for interpreting the results would improve the manuscript. Second, the reliance on self-reported measures for variables such as sleep duration, food practices, and screen time introduces potential bias. It might be useful to discuss how these biases could affect your findings or consider supplementing the questionnaire data with objective measures in future studies. Third, although the formation of latent variables for food practices is innovative, the rationale behind the grouping of certain items—particularly the separate treatment of vegetables and fruits—could be explained more thoroughly to clarify why these items did not load with other healthy eating indicators. Additionally, while the structural equation models appear robust, a more detailed presentation of the model fit indices and any modifications made based on the “modification indices” would enhance transparency.

In summary, your study provides valuable insights into the influence of school shift on adolescent health behaviors and nutritional status. Addressing these methodological and interpretative issues would further solidify the conclusions and offer a clearer pathway for future interventions.

6. PLOS authors have the option to publish the peer review history of their article (what does this mean? ). If published, this will include your full peer review and any attached files.

**Do you want your identity to be public for this peer review?** For information about this choice, including consent withdrawal, please see our Privacy Policy .

Reviewer #1: No

Reviewer #2: No

---

## [Author Response · Author response to Decision Letter 1]

22 Mar 2025

Dear Editor,

We sincerely appreciate the insightful comments and constructive feedback provided by the reviewers on our manuscript titled "Influence of study shift on the interrelationships among chronobiological factors, health practices, and anthropometry in adolescents". We carefully considered all suggestions and have made substantial revisions to address the concerns raised. Below, we provide a detailed response to each comment, outlining the modifications incorporated into the Revised Manuscript using Track Changes. Additionally, we made necessary adjustments to align with PLOS ONE's style requirements. These include:

- Replacing "Figure" with "Fig.";

- Renaming all supporting information files, changing "Supplementary Tables 1, 2, and 3" to "S1, S2, and S3 Tables";

- Removing supporting tables from the main manuscript and submitting them as separate files.

We appreciate the time and consideration given to our work and look forward to your further feedback and that of the reviewers.

Best regards,

The authors.

REVIEWER 1

1. In Table 2 there is text that is written in Portuguese language.

Response: Thank you for your feedback. We have corrected the text in Table 2, ensuring that all content is now presented in English as required.

2. Please develop the limitation section including the reliance on self-reported data (e.g., food practices, screen time, and sleep duration) that may introduce bias or inaccuracies.

Response: Thank you for your feedback. We have revised the limitations section to provide a more detailed discussion of the potential biases and inaccuracies associated with self-reported data. These modifications can be found between lines 506–517.

REVIEWER 2

1. The cross-sectional design limits the ability to infer causality, which you acknowledge; however, further discussion on this limitation and its implications for interpreting the results would improve the manuscript.

Response: Thank you for your valuable feedback. In response, we have expanded the discussion on the limitations of our cross-sectional study design, addressing its implications for causal inference and interpretation of results. These revisions can be found between lines 491–492 and 499–505.

2. The reliance on self-reported measures for variables such as sleep duration, food practices, and screen time introduces potential bias. It might be useful to discuss how these biases could affect your findings or consider supplementing the questionnaire data with objective measures in future studies.

Response: Thank you for your feedback. We have revised the limitations section to further elaborate on the potential biases associated with self-reported data and their possible impact on our findings. Additionally, we have acknowledged the value of incorporating objective measurement tools in future research. These adjustments can be found between lines 506–517.

3. Although the formation of latent variables for food practices is innovative, the rationale behind the grouping of certain items - particularly the separate treatment of vegetables and fruits - could be explained more thoroughly to clarify why these items did not load with other healthy eating indicators.

Response: Thank you for your valuable observation. We recognize the importance of clarifying the separation of vegetables and fruits from other healthy eating indicators in the analysis. To address this, we have incorporated an additional explanation between lines 123 and 129.

In our model, latent variables require at least three indicators. Although fruits and vegetables were initially assigned to the same factor in the exploratory factor analysis, they formed a group with only two variables, making it methodologically inadequate to create a latent variable. Therefore, they were analyzed separately in the structural equation model.

Forcing fruits and vegetables to load into Factor 1, which comprises breakfast, lunch, dinner, and beans, did not yield meaningful results, as they did not present satisfactory factor loadings (≥ 0.3) in the confirmatory factor analysis for either school shift. This suggests that these variables had low correlation with Factor 1 and did not adequately represent its theoretical construct.

A possible explanation for this result is the study's setting: the federal educational institution provides free, high-quality meals daily, which may contribute to lesser differentiation of fruit and vegetable consumption. Furthermore, unlike beans and main meals, fruits and vegetables may be consumed in different contexts, such as snacks, and their inclusion in main meals depends on individual habits and availability. This partly explains their allocation to Factor 3 instead of Factor 1. As a result, Factor 1 was renamed from "healthy food practices" to "complete meals," while Factor 2 remained "unhealthy foods," and fruits and vegetables were analyzed as independent variables.

4. While the structural equation models appear robust, a more detailed presentation of the model fit indices and any modifications made based on the “modification indices” would enhance transparency.

Response: We appreciate your feedback. To improve transparency regarding the modeling process, we have clarified the description of the modification indices in the Statistical Analysis section of Materials and Methods (lines 229-232) and provided a more detailed description of the model fit indices in the corresponding results table (lines 283-290). These revisions ensure a clearer presentation of the methodological decisions taken and the robustness of the structural equation models.

---

## [Editor Report · Decision Letter 1]

25 Mar 2025

Influence of study shift on the interrelationships among chronobiological factors, health practices, and anthropometry in adolescents

PONE-D-25-00055R1

Dear Dr. Cintia Chaves Curioni,

We’re pleased to inform you that your manuscript has been judged scientifically suitable for publication and will be formally accepted for publication once it meets all outstanding technical requirements.

Kind regards,

Miray Budak

Academic Editor

PLOS ONE

Additional Editor Comments (optional):

Dear Authors,

Thank you for your thorough and thoughtful revisions to your manuscript titled “Influence of study shift on the interrelationships among chronobiological factors, health practices, and anthropometry in adolescents.” Your responses effectively address the reviewers’ concerns, and the modifications have strengthened the clarity and rigor of your study.

Notably, the expanded discussion on study limitations, particularly regarding self-reported data and cross-sectional design, adds valuable context to your findings. Your detailed justification for the latent variable structure and factor loadings enhances the transparency of your methodology. Additionally, the clarification of model fit indices contributes to the robustness of your statistical analysis.

After reviewing the revised manuscript and your responses, I am pleased to inform you that I recommend acceptance of your manuscript for publication in PLOS ONE. Congratulations on this achievement, and we look forward to seeing your work published.

Best regards,

Miray Budak

---

## [Editor Report · Acceptance letter]

PONE-D-25-00055R1

PLOS ONE

Dear Dr. Curioni,

I'm pleased to inform you that your manuscript has been deemed suitable for publication in PLOS ONE. Congratulations! Your manuscript is now being handed over to our production team.

Kind regards,

on behalf of

Dr. Miray Budak

Academic Editor

PLOS ONE